# High atherogenic risk concomitant with elevated HbA1c among persons with type 2 diabetes mellitus in North Ethiopia

**Hagos Amare Gebreyesus**[1,2]*****, **Girmatsion Fisseha Abreha**[2], **Sintayehu Degu Besherae**[2], **Merhawit Atsbha Abera**[2], **Abraha Hailu Weldegerima**[2], **Aregawi Haileslassie Gidey**[2], **Afework Mulugeta Bezabih**[2], **Tefera Belachew Lemma**[1], **Tsinuel Girma Nigatu**[3,4]

**1** Department of Nutrition & Dietetics, Jimma University, Jimma, Ethiopia, **2** College of Health Sciences, Mekelle University, Mekelle, Ethiopia, **3** Department of pediatrics and child health, Jimma University, Jimma, Ethiopia, **4** Jimma University Clinical and Nutrition Research Center (JUCAN), Jimma University, Jimma, Ethiopia

* yom_hag@yahoo.com

**Data Availability Statement:** All relevant data are within the manuscript.

## Abstract

### Background

Type 2 diabetes mellitus (T2DM) is a chronic disease associated with worse clinical presentation. However, the current investigation practices in Ethiopia have limitations to demonstrate the scope of the clinical burden. Hence, this study was aimed at assessing the glycemic status and coronary heart disease (CHD) risk of persons with T2DM using HbA1c and atherogenic index of plasma (AIP).

### Method

This institution-based cross-sectional study was conducted among 421 adults with T2DM from September to November 2019. Demographic, socioeconomic, and lifestyle data were collected through a face-to-face interview. Clinical information was retrieved from medical records whereas anthropometric and biochemical measurements were performed using the WHO protocols. Glycemic status was determined using HbA1c and CHD risk assessed using an atherogenic index of plasma (AIP). Gaussian variables were expressed using mean and standard deviation (SD), Log-normal variables using geometric mean and 95% CI and non- Gaussian variables using median and interquartile ranges. Categorical variables were summarized using absolute frequencies and percentages. Multivariable logistic regression was used to identify factors associated with glycemic control with a statistical significance set at 5%.

### Result

A total of 195 male and 226 female subjects were involved in this study. The results demonstrated that 77% (324) had HbA1c value $\geq$7% and 87.2% (367) had high atherogenic risk for CHD. Besides, 57% and 67.9% of persons with T2DM had metabolic syndrome according to International Diabetes Federation (IDF) and the National Cholesterol Education Program

**Funding:** The author(s) received no specific funding for this work.

**Competing interests:** The authors have declared that no competing interests exist.

—Adult treatment panel III (NCEP-ATP III) criteria, respectively. About 36.8% had one or more comorbidities. Having healthy eating behavior [AOR 1.95; CI 1.11–3.43] and taking metformin [AOR 4.88; CI 1.91–12.44] were associated with better glycemic outcomes.

## Conclusion

High AIP level concomitant with poor glycemic control indicates increased risk for coronary heart disease among persons with T2DM in Northern Ethiopia.

## Introduction

Type 2 diabetes mellitus (T2DM) is a chronic disease associated with acute and long-term complications [1, 2]. Cardiovascular (CV) complication, largely coronary heart disease (CHD), is one of the significant complications that lead to premature death [1, 3]. Dyslipidemia, a key mediator of the T2DM-CHD link [4], is chiefly atherogenic and is characterized by elevated triglycerides (TGs), reduced HDL cholesterol levels, and raised small, dense LDL (sdLDL) particle levels [5].

T2DM can be considered a CHD risk equivalent. It is thus invaluable to monitor the risk level by assessing glycaemia and dyslipidemia to calibrate the intensity of management with risk. Achieving this goal, in turn, requires valid diagnostic and prognostic markers [6]. However, current Ethiopia practices rely on fasting serum glucose and conventional lipid profiles [7–11]. Fasting serum glucose is less reproducible and is inferior to predict long-term outcomes [12, 13].

Similarly, despite its merit to gauge the quantitative derangement of lipoproteins, the existing lipid profiling leaves the residual risk connected with qualitative LDL changes undetected [14, 15]. Therefore, the risk monitoring made using the approaches mentioned above could have less quality, and thus the clinical management as a consequence. Hence, it is crucial to consider a valid glycemic status metric and include an index of residual CHD risk attributed via the atherogenic sdLDL particles.

As an average value of the previous 8–12 weeks, HbA1c measurement is robust for biological variability and is well suited to predict the occurrence and progression of chronic complications [6, 12, 13]. Hence, it is used as a gold standard to assess glycemic status and diabetes-related outcomes [6]. Likewise, the atherogenic index of plasma (AIP) serves as a surrogate measure for the less feasible sdLDL particle assay and predicts CHD risk better [16]. It can also be easily computed from the standard lipid parameters without adding any extra cost [16]. therefore, the present cross-sectional study was conducted with the aim of evaluating the glycemic status and coronary heart disease risk of persons with T2DM using HbA1c and AIP. Accordingly, we found that significant proportion of them had poor glycemic control and increased risk for CHD.

## Methods and materials

### Study setting

The study was conducted in two general hospitals: Adigrat Hospital from the Eastern zone and Mekelle Hospital from the Mekelle especial zone of Tigray Region, North Ethiopia. Both hospitals serve as referral centers for health facilities in their respective zones including a stand-alone clinic for Non-Communicable Diseases (NCDs) like diabetes. A respective of 527 and

684 persons with DM were under follow-up in Adigrat and Mekelle Hospitals during the study period.

## Study design and participants

A facility-based cross-sectional study was conducted from September to November 2019. Participants were individuals 18 years and above with T2DM under clinical follow-up during the study period. Pregnant or breastfeeding women or individuals with documented cognitive impairment were excluded.

## Sample size and sampling technique

Using single population proportion formula, we calculated a sample size of 421, taking a 58.8% prevalence of poor glycemic control among persons with T2DM in Addis Ababa, Ethiopia [17], reliability coefficient Z = 1.96 at a 95% confidence interval, a margin of error (d) = 0.05 and 10% non-response rate. With proportional allocation, 237 and 184 participants were enrolled from Mekelle and Adigrat Hospitals, respectively, through systematic sampling using the hospital registry as a frame.

## Data collection

**Socio-demographic and lifestyle data.** Demographic, socioeconomic, and lifestyle data were collected through face-to-face interviews using a pretested structured questionnaire that was developed by the investigators. The socioeconomic questionnaire was adapted from Demographic Health Survey (DHS) [18]. The lifestyle questionnaire was adopted from the WHO stepwise approach (STEPS) instrument [19].

**Clinical information and anthropometry.** Clinical information was retrieved from medical records. Body weight was measured to the nearest 0.5 kg (Seca 755 with a stadiometer, Seca, Hamburg, Germany) and height to the nearest 0.5 cm (Seca 755 with a stadiometer, Seca, Hamburg, Germany) with the subjects positioned at the Frankfurt Plane and the four points (heel, calf, buttocks, and shoulder) touching the vertical stand. Waist and hip circumferences were measured using a stretch-resistant metric tape with subjects standing with their feet fairly close together and their weight equally distributed to each leg. Waist circumference was measured in the horizontal plane, midway between the iliac crest and the lower rib margin. Hip circumference was measured at the highest extension of the buttock. Both measurements were taken in duplicates, recorded to the nearest 0.5 cm. The waist-to-hip ratio (WHR) was derived from the average record dividing the WC (cm) by the HC (cm). Likewise, waist- to- height ratio (WhtR) was calculated via dividing WC (cm) by height in meter square ($m^2$). All the measurements were taken with participants in minimal clothing and barefoot [19, 20].

**Determination of physical activity status.** Physical activity of participants was collected as part of the lifestyle interview through Global Physical Activity Questionnaire (GPAQ) [21]. The total time spent in physical activity during a full week period including activity for work, during transport and leisure time (sports) and the intensity of the physical activities were accounted in the process. To calculate total physical activity metabolic equivalent minute (MET) values were applied to the time variables in the GPAQ data according to the intensity of the activity (moderate or vigorous). Four (4) and eight (8) MET values were applied for moderate and vigorous intensity activity, respectively. Accordingly, total physical activity in MET-minutes/week was calculated by multiplying the total numbers of minutes spend in moderate-intensity physical activity by 4 and total numbers of minutes spend in vigorous-intensity physical activity by 8.

**Blood pressure measurement.** Blood pressure was measured using a digital sphygmoma-nometer (OMRON, Brazil) with the arm held on a table at heart level. Duplicate measurements of systolic (SBP) and diastolic blood pressure (DBP) were taken at 5 minutes intervals after the participant has rested in a chair for at least 5 min. An average of the two readings was recorded in units of millimeter mercury [19].

**Biochemical measurements.** Whole blood and serum samples were taken after overnight fasting for biochemical analysis. About 6 ml of venous blood was collected into two vacutainer tubes; one with serum separator (SST™) and the other with EDTA anticoagulant. The blood drawn into an EDTA-containing tube was refrigerated at 4˚C and was subjected to HbA1c determination in Humameter A1C. The blood collected in the serum separator tube was allowed to clot for 30 minutes and centrifuged at 3500 rpm for 5 minutes to separate the serum from the formed elements. Right after that, serum samples were aseptically aliquoted into cryo-tubes using disposable rubber pipettes and stored at -70˚c until the time of analysis. Finally, serum samples were thawed and analyzed for lipid parameters including total cholesterol (TC), low-density lipoprotein cholesterol (LDL), high-density lipoprotein cholesterol (HDL), and triglycerides (TG). The analysis was carried out in pentra C 400 clinical chemistry analyzer using ABX Pentra C 400 reagents from Horiba Company. Lipid indices were calculated from the measured conventional lipid parameters. TC/ HDL ratio was calculated by dividing TC (mg/dl) to HDL (mg/dl). The atherogenic index of plasma (AIP) was calculated as a logarith-mic transformation of the ratio of TG to HDL (log TG/HDL) [22].

## Statistical analysis

Once checked for clarity and completeness of information data was entered into Epidata 3.1 (Xunta de Galicia, Spain & PAHO, USA) while cleaning and analysis were done using SPSS for windows version 23 (IBM Corp, New York). Wealth index was created as a proxy measure for socioeconomic status using Principal Component Analysis (PCA). The index in the first component was included in the logistic regression as a covariate to represent the wealth status of the participants. Physical activity status was determined as per the WHO protocol. Individuals who scored a minimum of 600 MET-minutes per week in moderate, vigorous or combined moderate and vigorous-intensity physical activities were categorized as physically active. While individuals scoring a total physical activity less than 600 MET minutes per week were considered insufficiently active.

Baseline data of participants were summarized using descriptive statistics. Normally distrib-uted variables were expressed using mean and standard deviation (SD), while non-normally distributed variables were log-transformed. Variables normalized with log-transformation were expressed using geometric mean and 95% CI whereas variables that remain non-Gauss-ian were described with median and interquartile ranges. Categorical variables were summa-rized using absolute frequencies and percentages. Bivariate logistic regression was used to model the association between potential risk factors and glycemic control status. Covariates with a p-value of 0.25 and less during the bivariate analysis and others assumed to have strong influence based on prior evidence were then included in the multivariable logistic model. The absence of considerable correlation among the covariates was asserted through the multicolli-nearity test. The level of statistical significance was set at 5% ($p < 0.05$).

## Classification criteria and operational definition

**Glycemic control.** According to ADA T2DM patients who manage to keep their HbA1c below 7.0% are considered to have good glycemic control while those with $\geq$ 7.0% are consid-ered to have poor glycemic control [23].

**IDF criteria for metabolic syndrome.** Since our participants were persons with T2DM they were classified as having MetS if central obesity (as defined by waist circumference ≥ 94 cm for men and ≥ 80 cm for women) co-exists with any one of the following components: Raised TG levels ≥150 mg/dl, or specific treatment for this lipid abnormality, reduced HDL-cholesterol < 40 mg/d in men and < 50 mg/dl in women, or specific treatment for this lipid abnormality and raised arterial blood pressure: systolic BP ≥ 130 or diastolic BP ≥ 85 mmHg or treatment of previously diagnosed hypertension [24].

**NCEP-ATP III criteria for metabolic syndrome.** Persons with T2DM were classified as having MetS if three or more of the following risk factors coexist: waist circumference > 102 cm for men or > 88 cm for women, serum triglyceride level ≥ 150 mg/dl, HDL cholesterol < 40 mg/dl in men or < 50 mg/dl in women, arterial blood pressure ≥ 130/85 mmHg and fasting plasma glucose ≥110 mg/dl [25].

**Treatment adherence.** Type 2 diabetes patients who take their medication without interruption and keep their timing as well are considered strongly adherent.

Type 2 diabetes patients who take their medication without interruption regardless of timing are considered moderately adherent.

Type 2 diabetes patients who interrupt their medication and fail to keep their timing are considered poorly adherent.

## Ethical considerations

This study was approved by the Institutional Review Board of the Institute of Health, Jimma University (IHRPGD/467/2018). Permission to conduct the study was acquired from Tigray Regional Health Bureau and authorities from the selected hospitals. Moreover, written informed consent was obtained from each participant and blood sample was collected on the basis of voluntariness. There was no significant harm incurred to the participants in connection with the volume of blood collected and the drawing process. We also managed to link participants having panic results with their physicians.

## Results

### Socio-demographic characteristics

A total of 421 persons with T2DM were enrolled in this study. Their mean age was 58.2 years (+/- 11). About 53.7%) (226) were females, 83.4% (351) were urban residents, 53.4% (225) were married and 12.6% (53) were single. The remaining 12.4% (52) and 21.6% (91) were divorced and widowed, respectively. About 97.4% (410) were ethnic Tigreans and the remaining 2.6% (11) being Amhara, Oromo, and Afar. Religious-wise, 91.9% (387) were Orthodox Christianity followers and 5% (31), 2.1% (9), and 1% (4) were Muslim, Catholic, and protestants, respectively. In terms of educational status, 43.9% (185) were illiterate, 29.5% (125) were able to read and write or had primary education and 26.4% (111) were educated to secondary level and beyond.

### Lifestyle and clinical features

Regarding lifestyle, 74.8% (315) were physically active and 98.1% (413) and 98.6% (415) were non-smokers and non-chat chewers, respectively (Table 1). All the participants were taking drugs for T2DM with a reported high, 387 (91.9%), adherence and a median (IQR) follow-up period of 5 (6) years. Moreover, they indicated that they use a combined diet and medication, 181(43%), approach as a means of controlling their sugar. However, their dietary self-care merely focuses on food selection [26]. About 36.8% (155) of them had one or more documented comorbidities of which hypertension was taking the lead, 101(24%).

**Table 1. Lifestyle and clinical characteristics of persons with T2DM in Northern Ethiopia (n = 421).**

| Variable | Category | Frequency (%) |
|---|---|---|
| Physical activity level | Physically active | 315 (74.8) |
| | Insufficiently active | 60 (14.3) |
| | Sedentary | 46 (10.9) |
| Cigarette smoking | Yes | 8 (1.9) |
| | No | 413 (98.1) |
| Khat chewing * | Yes | 6 (1.4) |
| | No | 415 (98.6) |
| Alcohol intake | Yes | 119 (28.3) |
| | No | 302 (71.7) |
| Alcohol intake frequency per month | 0–1 | 48 (11.4) |
| | 2–3 | 46 (10.9) |
| | 6–12 | 12 (2.9) |
| | 16 or more | 13 (3.1) |
| Average number of daily alcoholic drinks intake | 1–2 | 91(21.6) |
| | 3–4 | 25 (5.9) |
| | 5 and more | 3 (0.7) |
| Duration of follow-up at DM clinic | 1–5 years | 223 (53) |
| | 6–10 years | 132 (31.3) |
| | > 11 years | 66 (15.7) |
| Means of sugar control | Medication | 95 (22.6) |
| | Diet & medication | 181 (43) |
| | Exercise & medication | 24 (5.7) |
| | Diet, exercise & medication | 121 (28.7) |
| Type of medication for diabetes | Metformin | 115 (27.3) |
| | Glibenclamide | 69 (16.4) |
| | Metformin & Glibenclamide | 171 (40.6) |
| | Insulin | 62 (14.7) |
| | Insulin + Metformin | 4 (1.0) |
| Treatment adherence | High | 387 (91.9) |
| | Moderate | 30 (7.1) |
| | Poor | 4 (1) |
| Presence of comorbidity | Yes | 155 (36.8) |
| | No | 266 (63.2) |
| Type of comorbidity | Hypertension | 101 (24.0) |
| | Nephropathy | 16 (3.8) |
| | Neuropathy | 19 (4.5) |
| | Retinopathy | 38 (9) |
| | Cardiovascular | 9 (2.1) |
| | 2 or more comorbidities | 24 (5.7) |

*a plant used as a stimulant.

## Anthropometric and biochemical status

Table 2 describes anthropometric and biochemical measurements. The mean values of BMI, TG, TC, and median of LDL were below the standard cut-offs. Whereas, the median values of WHR, WHtR, and SBP were above the cut point. Similarly, the means of HbA1c, TC/HDL, and AIP were above the cut point in both sexes. While the mean for HDL was below the cut

**Table 2. Distribution of physical and biochemical measurements for persons with T2DM (n = 421).**

| Variable | | N | Mean/ Median | Classification | Frequency (%) |
|---|---|---|---|---|---|
| BMI, kg/m$^2$ | | 421 | 22.5 (22.2, 22.9)* | < 18.5 | 53 (12.6) |
| | | | | 18.5–24.9 | 250 (59.4) |
| | | | | ≥ 25 | 118 (28) |
| WC | M | 195 | 90.44 +/-12^ | < 94 cm | 112 (57.4) |
| | | | | ≥ 94 cm | 83 (42.6) |
| | F | 226 | 88.9 +/-12.1^ | < 80 cm | 53 (23.5) |
| | | | | ≥ 80 cm | 173 (76.5) |
| WHR | M | 195 | 0.94 (0.11)≠ | < 0.9 | 70 (35.9) |
| | | | | ≥ 0.9 | 125 (64.1) |
| | F | 226 | 0.89 (0.09)≠ | < 0.85 | 69 (30.5) |
| | | | | ≥ 0.85 | 157 (69.5) |
| WHtR | | 421 | 0.55 (0.54, 0.55)^ | < 0.5 | 102 (24.2) |
| | | | | ≥ 0.5 | 319 (75.8) |
| SBP | | 421 | 129 (23)≠ | < 130 mmHg | 212 (50.4) |
| | | | | ≥ 130 mmHg | 209 (49.6) |
| DBP | | 421 | 80 (20)≠ | < 85 mmHg | 289 (68.6) |
| | | | | ≥ 85 mmHg | 132 (31.4) |
| HgA1c | | 421 | 8.4 (8.2, 8.6)* | < 7% | 97 (23) |
| | | | | ≥ 7% | 324 (77) |
| Total cholesterol | | 421 | 180 (175.6, 184.3)* | < 200 mg/dl | 273 (64.8) |
| | | | | ≥ 200 mg/dl | 148 (35.2) |
| LDL | | 421 | 96.2(40.7)≠ | < 100 mg/dl | 240 (57) |
| | | | | ≥ 100 mg/dl | 181 (43) |
| HDL | M | 195 | 35.7 (34.4, 36.8)* | ≥ 40 mg/dl | 66 (33.8) |
| | | | | < 40 mg/dl | 129 (66.2) |
| | F | 226 | 38 (36.7, 39.4)* | ≥ 50 mg/dl | 41(18.1) |
| | | | | < 50 mg/dl | 185 (81.9) |
| Triglyceride | | 421 | 130.4(123.7, 137.4)* | < 150 mg/dl | 248 (58.9) |
| | | | | ≥ 150 mg/dl | 173 (41.1) |
| TC/HDL | | 421 | 4.87 (4.75, 5.0)* | < 0.5 | 218 (51.8) |
| | | | | > 0.5 | 203 (48.2) |
| AIP | | 421 | 0.55 (0.52, 0.58)^ | < 0.11 | 37 (8.8) |
| | | | | ≥ 0.11 | 384 (91.2) |

Abbreviations & symbols: CI, confidence interval IQR, interquartile range,

^ arithmetic mean+/-SD,

*Log transformed data and hence geometric mean (95% CI),

≠ median (IQR).

point. This may signify that ratios or indices are more sensitive in capturing biochemical derangements than absolute measurements.

Regarding the anthropometric status of the participants, 59.4% (250) had a normal body mass index. In contrast, 60.8 (256), 67 (282), and 75.8% (319) of which had an overall elevated WC, WHR, and WhtR, respectively. Gender-based proportions for WC and WHR are presented in Table 2. As indicated in the same table, a considerable proportion of participants had also problems in maintaining their glycemic status and lipid parameters within the standard limit. A total of 77% (324) participants had poor glycemic control as demonstrated with

HbA1c ≥ 7%. Likewise, 91% (383) had dyslipidemia in at least one parameter. Only 9% (38) had all four components normal while 12.4% (52) had all their values deranged. The remaining 30.9 (130), 29.7 (125) and 18.1% (76) had dyslipidemias in one, two, and three components, respectively. The most frequent type of single dyslipidemia was low HDL 74.6% (314), while in the case of mixed dyslipidemia; it was hypertriglyceridemia with low HDL 34% (143) followed by high LDL with low HDL 30.6% (129).

## Coronary heart disease risk status

Fig 1 displays the coronary heart disease risk level of participants as estimated using AIP and Table 3 outlines the prevalence of metabolic syndrome. The results showed that 8.8% (37) had low risk while 4% (17) and 87.2% (367) of participants had intermediate and high risk for coronary heart disease, respectively. Likewise, metabolic syndrome was found in 57% (240) and 67.9% (289) participants according to IDF and NCEP ATP III criteria, respectively. Both criteria revealed a significant burden of metabolic syndrome among women as compared to men (P <0.001). Concerning age, those within the 45–64 years category had more prevalence of metabolic syndrome on both criteria, not significant based on IDF though. However, there was no significant difference in the prevalence of metabolic syndrome with glycemic status on either criterion.

## Factors associated glycemic control

Table 4 shows results for bivariate and multivariable logistic regression analysis of factors associated with the glycemic status of the participants. In the bivariate model age at diagnosis, educational status, wealth index, duration of diabetes, treatment type, and eating behavior were significantly associated with glycemic status at the level of $\alpha$ = 25%. All these variables and others including physical activity level, means of sugar control, waist to height ratio (WhtR), and presence or absence of concordant comorbidities were taken into the multivariable model. The results indicated that only eating behavior, treatment type, and wealth index were able to retain their statistically significant association at the level of $\alpha$ = 5%. Participants having partial healthy eating behavior [AOR 1.95; CI 1.11–3.43] and taking Metformin [AOR 4.88; CI 1.91–12.44] as a treatment had good glycemic control (HbA1c <7%). On the other hand, a unit increase in wealth index was associated with a 1% decrease in achieving glycemic target [AOR 0.99; CI 0.98–0.99].

## Discussion

This study was designed to assess the glycemic status and cardiovascular risk of persons with T2DM in Northern Ethiopia. The results indicated that 77% had poor glycemic control and 87.2% had high atherogenic risk for CHD. Besides, 57 and 67.9% of persons with T2DM had metabolic syndrome according to IDF and NCEP-ATP III criteria, respectively. Eating behavior and treatment type demonstrated a statistically significant association with the key indicator i.e HbA1c status.

Shreds of evidence show that strict glycemic control is crucial to prevent diabetes-related complications and hence reduce concomitant hospitalization and mortality [26]. On the other hand, poorly-controlled glycemia increases the likelihood of diabetic complications [27]. According to ADA, diabetics who failed to keep their HbA1c below 7.0% are considered to have poor glycemic control [28]. In this study poor glycemic control was observed in 77% of persons with T2DM. This is comparable with 70.8–81.9% [29–31] but higher than 48.7%–57.5% [7, 32–34] prevalence of poor glycemic control revealed in other Ethiopian settings. Again it is higher than the global estimate [27] and the rate in high-income countries [35, 36].

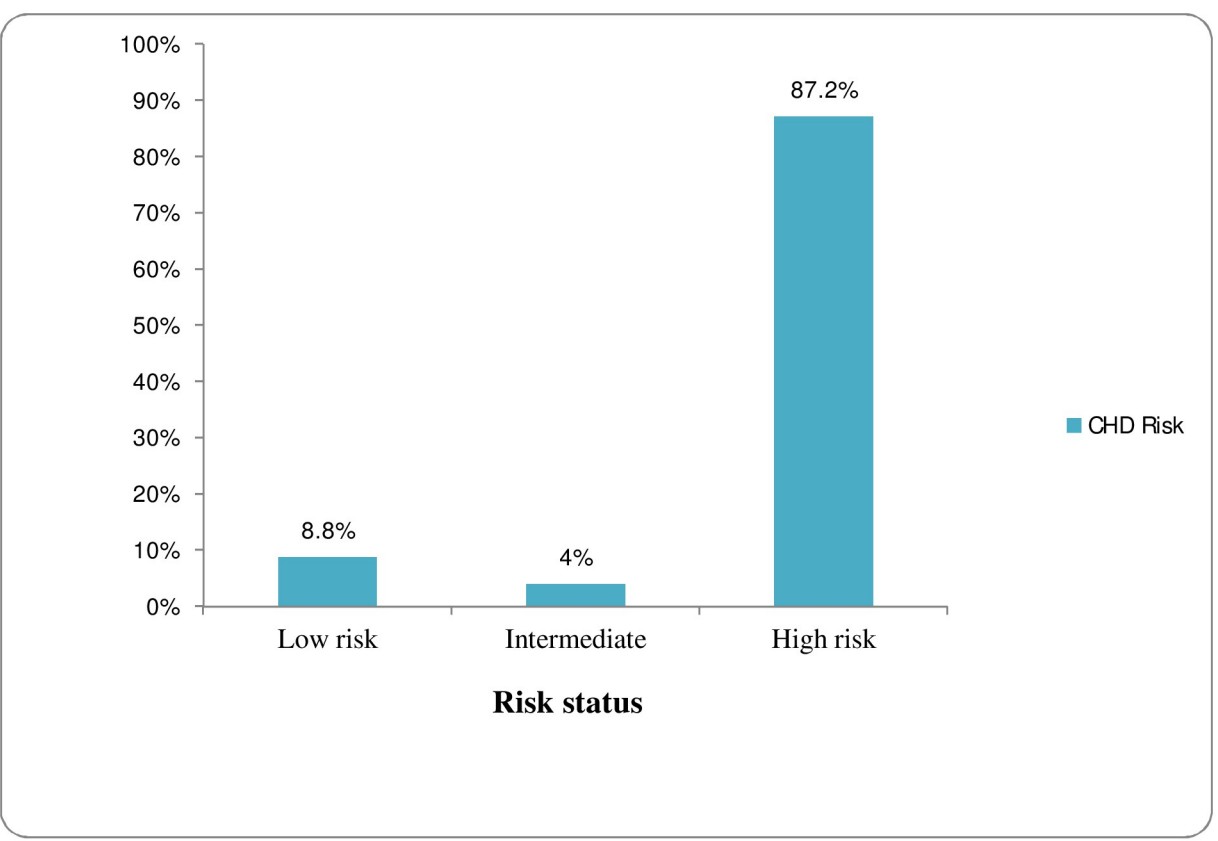

**Fig 1. Coronary heart disease risk as predicted by AIP.** Note: low risk (AIP <0.11), intermediate (AIP, 0.11–0.21) & high risk (AIP >0.21).

However, it is consistent with the poor glycemic control prevalence of, 73.7–81.8% indicated in low-income countries [37–39]. The likely reasons for the observed variation could be differences in the knowledge on glycemic control, income, treatment modality, and clinical characteristics specific to each patient. Moreover, the unmatched increase in low-income countries could be attributable of a swift nutrition transition from fiber-rich traditional foods to calorie-

**Table 3. Proportion of metabolic syndrome (MetS) by sex, age, and HbA1c level (n = 421).**

| Features | | Classification criteria | | | | | |
|---|---|---|---|---|---|---|---|
| | | IDF | | | NCEP ATP III | | |
| Variable | Category | No MetS | MetS | P.value | No MetS | MetS | P. value |
| Sex | Male | 119 (65.7) | 76 (31.7) | <0.001 | 83 (61.5) | 112 (39.1)^ | <0.001 |
| | Female | 62 (34.3) | 164 (68.3) | | 52 (38.5) | 174 (60.8) | |
| Age category | 26–44 | 24 (13.3) | 26(10.8) | 0.626 | 23 (17) | 27(9.4) | 0.008 |
| | 45–64 | 95 (52.5) | 136 (56.7) | | 79 (58.5) | 152 (53.1) | |
| | 65+ | 62 (34.3) | 107 (32.5) | | 33 (24.4) | 107 (37.4) | |
| HbA1C | < 7% | 144 (79.6) | 78 (75) | 0.294 | 107 (79.3) | 217 (75.9) | 0.460 |
| | ≥ 7% | 37 (20.4) | 60 (25) | | 28 (20.7) | 69 (24.1) | |

^- frequency (%),

ADA—American Diabetic Association, IDF—International Diabetes Federation, NCEP ATP III-National cholesterol education program adult treatment panel III.

**Table 4. Factors associated with the glycemic status of persons with T2DM in North Ethiopia (n = 421).**

| Characteristics | Category | Crude odds ratio | | Adjusted odds ratio | |
|---|---|---|---|---|---|
| | | COR | 95% CI | AOR | 95% CI |
| Current age in year | 26–44 | 1.05 | 0.50–2.21 | | |
| | 45–64 | 0.81 | 0.49–1.33 | | |
| | 65+ | 1.00 | | | |
| Age at diagnosis | 21–40 | 0.71 | 0.35–1.44 | 1.32 | 0.52–3.35 |
| | 41–60 | 0.63 | 0.37–1.07 | 0.75 | 0.38–1.47 |
| | 61+ | 1.00 | | 1.00 | |
| Sex | Male | 1.12 | 0.71–1.76 | | |
| | Female | 1.00 | | | |
| Residence | Urban | 1.12 | 0.60–2.08 | | |
| | Rural | 1.00 | | | |
| Marital status | Single | 1.07 | 0.52–2.19 | | |
| | Married | 079 | 0.48–1.29 | | |
| | Divorced/Widowed | 1.00 | | | |
| Educational status | Illiterate | 1.65 | 0.90–2.99 | 1.62 | 0.78–3.35 |
| | Informal/primary | 1.60 | 0.84–3.03 | 1.55 | 0.72–3.33 |
| | Secondary & above | 1.00 | | 1.00 | |
| Wealth index | | 0.99 | 0.98–0.99 | 0.99* | 0.98–0.99 |
| Duration of DM | 0–4 years | 1.97 | 0.94–4.11 | 1.54 | 0.62–3.84 |
| | 5–10 years | 1.58 | 0.72–3.47 | 1.57 | 0.62–4.00 |
| | ≥ 11 years | 1.00 | | 1.00 | |
| Methods of sugar control | Medication | 0.88 | 0.47–1.64 | 1.24 | 0.59–2.64 |
| | Diet & Medication | 1.03 | 0.62–1.73 | 0.81 | 0.43–1.53 |
| | Diet, execs & med | 1.00 | | 1.00 | |
| Type of medication | Metformin | 3.64 | 1.64–8.10 | 4.88* | 1.91–12.44 |
| | Glebinclamide | 2.07 | 0.85–5.05 | 2.21 | 0.80–6.11 |
| | Metform & GLb | 1.29 | 0.58–2.90 | 1.39 | 0.56–3.46 |
| | Insulin | 1.00 | | 1.00 | |
| Treatment adherence | High | 1.17 | 0.49–2.77 | | |
| | Moderate | 1.00 | | | |
| Physical activity level | Physically active | 1.26 | 0.54–2.74 | 1.21 | 0.47–3.10 |
| | Insufficiently active | 1.25 | 0.49–3.21 | 1.14 | 0.38–3.45 |
| | Sedentary | 1.00 | | 1.00 | |
| Eating behavior | Partially healthy | 1.79 | 1.13–2.83 | 1.95* | 1.11–3.43 |
| | Unhealthy | 1.00 | | 1.00 | |
| Waist-to-height ratio | < 0.5 | 1.31 | 0.75–2.27 | 0.94 | 0.48–1.85 |
| | ≥ 0.5 | 1.00 | | 1.00 | |
| Comorbidity | Yes | 1.21 | 0.76–1.92 | 1.08 | 0.60–1.95 |
| | No | 1.00 | | 1.00 | |

*$P < 0.01$.

dense processed foods. Thus, facility and policy level efforts must be exerted to counter such adverse effects of feeding style right before overwhelming health and economic impacts happen.

Hyperglycemia and dyslipidemia generally co-exist in persons with T2DM manifesting poor glycemic control and their interaction increase the risk of vascular complications [40]. In

the present study, 91% of persons with T2DM had dyslipidemia in at least one parameter. Out of these 60% had mixed dyslipidemia with 12.4% of them having all their components deranged. This finding is higher than the prevalence of overall dyslipidemia revealed in a similar study from South Ethiopia [41]. In that study, 65.6% of participants had dyslipidemia of which 48.2% were with a mixed type. However, an almost equal proportion of their participants, 11.6%, had all their components deranged like in our study. Likewise, the current result is higher than the 63.8% and 60.5–70% dyslipidemias observed in Senegal and Nigeria, respectively [42–44]. The 46.3% mixed dyslipidemia detected in Northwestern Nigeria is also lower than this study but somehow equivalent to that of Southern Ethiopia. However, the existing finding is in agreement with findings of other studies which have shown 81.1–94% dyslipidemia and as high as 57.3–75% mixed occurrence over the globe [37, 45–49]. Despite such high prevalence dyslipidemia is quite amendable. Therefore, health care providers should perform routine screening and encourage their patients' modify their lifestyle, control their glucose, and initiate, or intensify lipid-lowering treatment when warranted.

Low HDL-C was the most common form of single dyslipidemia observed in 74.6% of persons with T2DM. And hypertriglyceridemia with low HDL-C was the most frequent combined type (34%). In tandem with this finding, the highest frequency of low HDL was also apparent in other studies [46, 48]. In the United Arab Emirates, low HDL was evident in 60% [48]. Similarly, it was seen amidst 69.11% of participants in Algeria [46]. Whereas, in Bangladesh hypertriglyceridemia appeared to be the most frequent type of single dyslipidemia, 60.7% [49]. Regarding the combined type, hypertriglyceridemia with low HDL-C observed in this study is consistent with that of 35.4% indicated in Bangladesh [49]. It is also comparable with the 41% shown in Nigeria [50]. This phenomenon confirms the notion that the pattern of diabetic dyslipidemia is more atherogenic than other types of dyslipidemia [51].

Reinforcing the aforementioned view, an AIP value (>0.11) demonstrating an intermediate or high-risk level of CHD was found in 91.2% of participants in this study. Again, this is in agreement with an increased risk level of AIP detected amongst 99.3% of persons with T2DM in Bangladesh [49]. The most likely cause of atherogenic dyslipidemia in T2DM is increased hepatic TG synthesis secondary to insulin resistance mediated free fatty acid flux. The increased triglyceride in turn modifies the circulating HDL and LDL-C leading to an atherogenic triad of hypertriglyceridemia, low HDL-C, and increased concentration of sdLDL particles [50, 52]. Such atherogenic triad establishes a proatherosclerotic milieu in the plasma and if it is not timely intervened will end up with premature atherosclerosis. Hence, it is worth targeting a high concentration of TGs for therapy as a means of curving this devastating complication.

Poor glycemic control and dyslipidemia are independent risk factors for CHD. Hence, T2DM individuals with elevated HbA1c and dyslipidemia can be considered as a very high-risk group for CHD. Substantiating this assumption more than half of the persons with T2DM in the current study had metabolic syndrome, which itself is an independent clinical marker for cardiac morbidity and mortality. The 57% prevalence of MetS revealed using IDF criterion is exactly matching with a study finding from Ethiopia, 57% [11], and that of a systematic review in Sub-Saharan Africa, 57.15% [53]. However, it is higher than the 51.1% demonstrated from another setting of Ethiopia [10] and lower than rates of 63.6, 64.9, and 66.8% observed in Nigeria, Iran, and Nepal, respectively [54–56]. On the other hand, the 67.9% prevalence of MetS obtained using the NCEP criterion is comparable with the rates of 70.1 and 70.3% both from Ethiopia [11, 57] and 64.8% indicated on a systematic review in Sub-Saharan Africa [53]. Nonetheless, it is higher than the 45.9 and 59.4% reported rates from Ethiopia [58, 59] and 58% from Ghana [60]. Conversely, it is lower than the occurrences of MetS among 73.4, 73.9, 75.6, and 96% Iranian, Nepalese, Caucasian, and Indians with T2DM, respectively [55, 56, 61, 62].

Ethnic and cultural variations may have a great role in the observed discrepancies in the prevalence of metabolic syndrome between Ethiopians and people from other nations [63]. However, the inconsistencies do exist even among Ethiopians. Hence, the variation could also arise from differences in age, gender composition, economic status, glycemic status, duration of T2DM, and largely lifestyle. For example, in this study significantly higher prevalence of MetS was found among women as compared to men. The increased prevalence of MetS among women could be attributable to hormonal oral contraceptive uses that can decrease the sensitivity of muscles to insulin which in turn can lead to impaired glucose metabolism and dyslipidemia [64]. It could also be due to menopause induced changes in body fat distribution precipitating its abdominal accumulation [65]. Potentiating this assumption, more women had more abdominal obesity as evidenced by waist circumference than men in this study.

Poor glycemic control is assumed to be the main perpetrator for most of the aforementioned clinical problems occurring among persons with T2DM [27]. Hence, optimization of glycemic control is crucial to curb the rising complications and alleviate the staggering health and economic impacts linked with [23, 66]. This necessitates the identification of factors that need to be targeted for intervention. One of the factors found to be associated with good glycemic control in this study was healthy eating behavior. The odds of good glycemic control was two times higher in those having healthy eating behavior as compared to persons with T2DM following unhealthy eating behavior. The favorable effect of a healthy diet demonstrated in this study corroborates the ADA's view stating healthy diet helps the achievement of glycemic targets and maintenance of proper weight which together help reduce complications and improve patients' quality of life [23]. This is chiefly because a healthy diet that favors eating whole grains, non-starchy vegetables, fruits, legumes, seafood, lean meats, low-fat dairy products, and vegetable oils and avoids trans-fats, refined carbohydrates, and sugar-sweetened beverages is rich in fiber and low in its glycemic load. Likewise, the relevance of healthy eating for better glycemic control was shown by other researchers [67, 68].

Treatment type also influences the outcome of glycemic control. In this study, better glycemic control was found among participants who used Metformin only. The likelihood of having good glycemic status was almost 5 times higher among persons with T2DM taking Metformin as compared to insulin. This may be because Metformin has good antihyperglycemic efficacy at a lower risk of hypoglycemia and hence encourages compliance to treatment [69]. Besides, higher odds of having good glycemic control was also observed among persons with T2DM who were taking Glebinclamide or a combination of metformin and glebinclamide relative to insulin, statistically insignificant though. Similarly, best glycemic control was attained on people taking monotherapy, followed by a combination of oral hypoglycemic agents (OHAs) as compared with those getting a combination of insulin and OHAs in most other studies [8, 70–72]. The variation could be explained by the difficulty of taking more drugs or ineffective insulin use related to inconvenience with storage and/or injection [73, 74]. Hence, efforts should be made to retain the patients on monotherapy.

## Conclusions

This study revealed that a significant proportion of persons with T2DM in North Ethiopia had substantial coronary heart disease (CHD) risk concomitant with poor glycemic control. From diagnostic rationale it has used HbA1c which is superior to fasting serum glucose in terms of accuracy and independent tracking of CHD risk. Substantiating its strength the median LDL-C in this study was within the normal range whereas the mean AIP was fivefold higher than the cutoff. This implies that atherogenic dyslipidemia continues to contribute to CHD risk even when LDL-C is set at target. This in turn underscores the need for targeting

atherogenic dyslipidemia on top of stringent LDL-C goal. Hence, estimating the residual risk of CHD connected with atherogenic dyslipidemia using AIP was a viable approach. In other words, making AIP assessment part and parcel of the routine lipid profile would have remarkable implications in improving the clinical management and epidemiologic prediction.

Healthy eating behavior and metformin monotherapy were associated with good glycemic outcomes. Hence, policymakers should promote dietary counseling and preparation of dietary guidelines contextualized to the setting of the patients. Moreover, frontline health care providers should counsel their patients to integrate healthy eating with proper medication use. This would prevent a rapid transition of patients from one to the next level of a regimen. As a shortcoming, the AIP values of the participants were not compared with their angiographic findings. Therefore, building on the findings of the current study, further research is warranted to correlate AIP with angiographically confirmed CHD.

## Supporting information

**S1 Appendix. English version of data collection tool.**
(DOCX)

**S2 Appendix. Tigrigna version of data collection tool.**
(DOCX)

## Acknowledgments

We would like to express our sincere appreciation to all study participants, data collectors, Ayder comprehensive specialized hospital for permitting us to use the laboratory facility, Mekelle University, and Jimma University for giving us the opportunity to conduct this study.

## Author Contributions

**Conceptualization:** Hagos Amare Gebreyesus, Tsinuel Girma Nigatu.

**Data curation:** Hagos Amare Gebreyesus, Sintayehu Degu Besherae.

**Formal analysis:** Hagos Amare Gebreyesus.

**Funding acquisition:** Hagos Amare Gebreyesus, Girmatsion Fisseha Abreha, Sintayehu Degu Besherae, Merhawit Atsbha Abera, Abraha Hailu Weldegerima.

**Investigation:** Hagos Amare Gebreyesus, Aregawi Haileslassie Gidey.

**Methodology:** Hagos Amare Gebreyesus, Tefera Belachew Lemma, Tsinuel Girma Nigatu.

**Supervision:** Afework Mulugeta Bezabih, Tefera Belachew Lemma, Tsinuel Girma Nigatu.

**Writing – original draft:** Hagos Amare Gebreyesus.

**Writing – review & editing:** Girmatsion Fisseha Abreha, Merhawit Atsbha Abera, Abraha Hailu Weldegerima, Afework Mulugeta Bezabih, Tefera Belachew Lemma, Tsinuel Girma Nigatu.

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
