## [Decision Letter · Decision Letter 0]

19 Jul 2021

PONE-D-21-17486

High atherogenic risk concomitant with elevated HbA1c among persons with type 2 diabetes mellitus in North Ethiopia: A cross-sectional study

PLOS ONE

Dear Dr. Gebreyesus,

Thank you for submitting your manuscript to PLOS ONE. After careful consideration, we feel that it has merit but does not fully meet PLOS ONE’s publication criteria as it currently stands. Therefore, we invite you to submit a revised version of the manuscript that addresses the points raised during the review process.

We look forward to receiving your revised manuscript.

Kind regards,

Xiao-Feng Yang, MD, PhD, FAHA

Academic Editor

PLOS ONE

Journal Requirements:

2. Please include additional information regarding the survey or questionnaire used in the study and ensure that you have provided sufficient details that others could replicate the analyses. For instance, if you developed the survey or questionnaire as part of this study and it is not under a copyright more restrictive than CC-BY, please include a copy, in both the original language and English, as Supporting Information. If the questionnaire is published, please provide a citation to the (a) questionnaire and/or (b) original publication associated with the questionnaire

Reviewers' comments:

Reviewer's Responses to Questions

**Comments to the Author**

1. Is the manuscript technically sound, and do the data support the conclusions?

Reviewer #1: Partly

Reviewer #2: Partly

2. Has the statistical analysis been performed appropriately and rigorously? 

Reviewer #1: Yes

Reviewer #2: I Don't Know

3. Have the authors made all data underlying the findings in their manuscript fully available?

Reviewer #1: Yes

Reviewer #2: Yes

4. Is the manuscript presented in an intelligible fashion and written in standard English?

Reviewer #1: Yes

Reviewer #2: Yes

5. Review Comments to the Author

Reviewer #1: - We prefer if you use the third person singular, instead of the first person singular or plural (e.g. "we").

- The major defect of this study is the debate or Argument is not clear stated in the introduction session. Hence, the contribution is weak in this manuscript. I would suggest the author to enhance your theoretical discussion and arrives your debate or argument.

- More suitable title should be selected for the article. Title should decrease to 10-12 words.

- The abstract should state briefly the purpose of the research, the principal results and major conclusions. An abstract is often presented separately from the article, so it must be able to stand alone.

- It is suggested to present the structure of the article at the end of the introduction.

- It is suggested to compare the results of the present research with some similar studies which is done before.

- More suitable title should be selected for the table 3 instead of “Distribution of metabolic syndrome (MetS) by sex, age, and glycemic status based on classification criteria (n=421).”.

- It is suggested to add articles entitled “D. Serwaa et al. Prevalence and Determinant of Erectile Dysfunction in Type II Diabetes Mellitus and Healthy Men”, “D. R. Paudel. Catastrophic Health Expenditure: An Experience from Health Insurance Program in Nepal” and “Phuoc-Tan Diep. Oxytocin May be Superior to Gliptins as a Potential Treatment for Diabetic COVID-19 Patients” to the literature review.

- Page 9: the following paragraph is unclear, so please reorganize that:

“Permission was acquired from Tigray Regional Health Bureau and participating institutions (Adigrat and Mekelle General Hospitals). Each participant provided informed consent and voluntarily gave a blood sample. There was no significant harm in connection with the volume of blood collected and the collection process. Participants with panic results were immediately linked to their physician.”

- Much more explanations and interpretations must be added for the Results, which are not enough.

- “Notation” should be added to the article.

- DOI of the references must be added (you can use “" ext-link-type="uri" xlink:type="simple">https://crossref.org/").

- Please make sure your conclusions' section underscore the scientific value added of your paper, and/or the applicability of your findings/results, as indicated previously. Please revise your conclusion part into more details. Basically, you should enhance your contributions, limitations, underscore the scientific value added of your paper, and/or the applicability of your findings/results and future study in this session.

Reviewer #2: The authors ran a cross sectional study of 421 T2DM participants with the purpose to assess the glycemic status and coronary heart disease (CHD) risk using HbA1c and atherogenic index of plasma. Their findings indicate that High AIP along with poor glycemic control as assessed by HbA1c, is associated with increased of CHD among diabetic patients. The study in general carry’s some novelty and clinical significance and it’s been designed well and sound. My comments:

Abstract: AIP, wasn’t defined when first mentioned.

Methods: line 142, the assessment of physical activity in the study wasn’t clearly described. Please do. Other than that, the methods were sound and presented in detail.

Results: The results are mostly limited to descriptive analysis, it’s not clear that is delivers exactly the message in the conclusions. The authors didn’t exactly assess the risk of CHD, as they stated in the abstract, they didn’t show the results or the statistical tools they used to assess that. Complete reformatting of the results section and running the required testes to clearly prove their conclusions is required.

6. PLOS authors have the option to publish the peer review history of their article (what does this mean?). If published, this will include your full peer review and any attached files.

Reviewer #1: No

Reviewer #2: No

---

## [Author Response · Author response to Decision Letter 0]

7 Sep 2021

Response to Reviewers

Dear esteemed editor and reviewers we heavily thank for your time and expertise; here follows is our response to your comments and suggestions and the manuscript is revised accordingly. 

Editor comments

1. Please ensure that your manuscript meets PLOS ONE’s style requirements, including those for file naming.

- We tried to review the templates and accordingly customize our manuscript to fit into the requirements. 

2. Please include additional information regarding the survey or questionnaire used in the study and ensure that you have provided sufficient details that other could replicate the analysis. For instance, if you developed the survey or questionnaire as part of this study and it is not under a copy right more restrictive than CC-BY, please include a copy, in both original language and English, as supporting information. If the questionnaire is published, please provide a citation to the (a) questionnaire and /or (b) original publication associated with the questionnaire.

- The tool used to assess demographic background, diabetic follow and nutrition information as well as the anthropometric and biochemical data of the participants is included as supporting information both in the original language and English versions. Moreover, references for other tools utilized in the study are cited under the data collection section on page 6, line 103-104.

3. PLOS requires an ORCID iD for the corresponding author in Editorial Manager on papers submitted after December 6th, 2016. Please ensure that you have an ORCID iD and that it is validated in Editorial manager. 

- The corresponding author has already a pre-existing ORCID iD and is authenticated in Editorial Manager following the given direction. 

Reviewer ♯1 comments

We prefer if you use the third person singular, instead of the first person singular or plural (e.g. “we”). 

- Thanks for your suggestion; however, we followed same approach. We are glad to amend if there is any that you could specify.

The major defect of this study is the debate or argument is not clearly stated in the introduction section. Hence, the contribution is weak in this manuscript. I would suggest the authors to enhance your theoretical discussion and arrives your debate or argument. 

- Thanks for the remark. Being a metabolic problem by itself, T2DM is also a formidable risk factor for cardiovascular problems, chiefly coronary heart disease (CHD). The attributable risk of T2DM for CHD equates with the level of glycemia. Hence, integrating periodic monitoring of glycemic status with the management process is critical to reduce or avert the CHD risk. Existing practices in Ethiopia utilize fasting serum glucose as a means of monitoring glycemic status. As a diagnostic tool fasting serum glucose measurement indicates the subject’s sugar level at the time of blood sampling. However, blood glucose concentrations are continually modified by various factors like diet, exercise and stress. Hence, fasting glucose measurement is subjected to substantial influence from day-to-day variations and hence, lacks reproducibility. In contrast, the concentration of HbA1c in the blood reflects the average glucose over the preceding 2-3 months and is robust for short-term variability. Therefore, HbA1c is more accurate and tracks the glycemic outcomes of persons with T2DM across time much better than the fasting glucose measurement. And this argument to use HbA1c over the less reproducible and predictive fasting serum glucose is portrayed in the introduction in a concise way. 

- In addition, sizeable risk of CHD attributable of T2DM is mediated by dyslipidemia. Hence, periodic monitoring of dyslipidemia among persons with T2DM is an established practice in the management of persons with T2DM. This is decisive to gauge the risk of CHD and caliber the management intensity accordingly. In the Ethiopian setting, the monitoring of lipid derangement among persons with T2DM is carried out by quantitative measurement of lipid parameters (routine lipid profile). Among these, LDL-C is considered to be the primary target for therapy. However, even with reducing LDL-C to the recommended level a substantial risk of coronary heart disease remains unfixed. This residual risk is connected with hypertriglyceridemia induced remodeling of HDL and LDL. Such qualitative alterations are characterized by a decrease in HDL size and enhancement of its renal clearance and hence loss of anti-atherogenic function. 

- On the other hand, the remodeling of LDL-C creates a preponderance of small, dense LDL (sdLDL) particles that are highly susceptible to oxidation and more atherogenic than their precursors. Overall, the qualitative alteration ends up in atherogenic dyslipidemia. However, assessing lipid levels through the routine lipid profile ignores the attribution from the qualitative dyslipidemia. Lipoprotein sub-fractionation of LDL particles is the best way to measure the residual risk attributed by sdLDL particles. However, this is financially and technically less feasible in a developing setting. Fortunately, atherogenic index of plasma (AIP) calculated as the ratio of log (TG/HDL) correlates well with the particle size and composition and serves as a surrogate measure for the less feasible sdLDL particle assay. And our intention to use AIP which is readily calculated from the routine lipid profile and account for the residual risk of CHD attributed by qualitative dyslipidemia is briefly described in the introduction. 

More suitable title should be selected for the article. Title should decrease to 10-12 words

- Suggestion accepted and correction is made as indicated in track change (page 1, line 1-3). For your information PLOS ONE guideline allows up to 250 characters but in our case the title contains less than 100 characters. 

The abstract should state briefly the purpose of the research, the principal results and major conclusions. An abstract is often presented separately from the article, so it must be stand alone. 

- Comment accepted and the introduction and result sections of the abstract are modified as indicated in track changes (abstract section on page 2, line 20-21 and line 34-35). Regarding the conclusion, the aim of the study was to assess glycemic status and CHD risk using HbA1c and AIP, respectively. Hence, we found high AIP level together with poor glycemic control signifying that the participants have an increased risk for CHD. This is parsimoniously presented as we have to comply with the word limitation stated for the abstract by the journal.

It is suggested to present the structure of the article at the end of the introduction

- Comment accepted and correction made as indicated in track changes (page 4, line 72-75).

It is suggested to compare the results of the present research with some similar studies which is done before

- Thanks for your suggestion. As can be confirmed from the discussion section we tried to compare our findings with other similar studies from the literature. For instance our finding on glycemic status (HbA1c) is compared with related articles from the literature in the discussion section on page 20, line 171-175. Likewise, we tried to follow similar manner for the remaining findings unless we faced dearth of relevant articles. 

More suitable title should be selected for the table 3 instead of “Distribution of metabolic syndrome (MetS) by sex, age and glycemic status based on classification criteria (n=421).

- Comment accepted and the title is modified as indicated in track changes in the result section page 19 line 271.

It is suggested to add articles entitled “D.Serwaa et al. Prevalence and Determinants of Erectile Dysfunction in Type II diabetes mellitus and healthy men”; “D.R. Paudel. Catastrophic Health Expenditure: An Experience from Health Insurance Program in Nepal” and “Phuoc-Tan Diep. Oxytocin may be Superior to Gliptins as a Potential Treatment for Diabetic COVID-19 patients” to the literature review. 

- Suggestion considered and included in where they fit in the paper as reference number 2 and number 67. 

Page 9: the following paragraph is unclear, so please reorganize that: 

- Comment duly accepted and reorganized as indicated in track changes under the methods section on page 10-11, line 207-212. 

Much more explanations and interpretations must be added for the results, which are not enough. 

- Comment accepted and explanations are added in the result section as indicated in track changes under the result section page 12, line 228-232; page 15, 240-241; page 18, 259-262.

“Notation” should be added to the article

- Comment accepted and a notations is added to table 4 and fig 1

DOI of the references must be added (you can use “http://crossref.org/”)

- Comment accepted and DOI of the references is included 

Please make sure your conclusions section underscores the scientific value added of your paper, and/or the applicability of your findings/results, as indicated previously. Please revise your conclusion part into more details. Basically, you should enhance your contributions, limitations, underscore the scientific value added of your paper, and/or the applicability of your findings/results and the future study in this session. 

- Thanks for your important insights and the conclusion section is restructured as indicated in track changes under the conclusion section on page 416-435.

Reviewer ♯2 comments

Abstract: AIP wasn’t defined when first used

- Comment accepted and correction made as indicated in track changes under the abstract section on page 2, line 22-23.

Methods: Line 142, the assessment of physical activity in the study wasn’t clearly described. Please do.

- Comment accepted and the information on how physical activity is assessed is indicated in track changes under the methods section on page 6-7, line 119-229 and page 8, line 159-163.

Results: The results are mostly limited to descriptive analysis. It is not clear that is delivers the message in the conclusion. The author didn’t exactly assess the risk of CHD, as they stated in the abstract, they didn’t show the results of the statistical tools they used to assess that. Complete reformation of the results section and running the required tests to clearly prove their conclusion is required.

- Thanks for the meticulous view. As described under the introduction section on page 4, line 68-70 and methods section on page 8, line number 150-151, the coronary heart disease risk was assessed using atherogenic index of plasma (AIP) as a surrogate marker of CHD risk. And AIP is computed taking the logarithmic transformation of the ratio of triglyceride to HDL-C, (log TG/HDL), which are components of the routine lipid profile. The cutoff value for AIP is 0.11. Hence, as described in the result section on page 16 and figure 1, the participants were classified in to three risk categories based on their AIP values. Persons with T2DM who had AIP value 0.11 were classified as having low risk. Whereas, persons with T2DM who had AIP level 0.11 to 0.21, and AIP level 0.21 were classified in to intermediate and high risk categories, respectively. Therefore, the participants CHD risk was assessed using AIP as a surrogate marker and were classified as having low, intermediate and high risk accordingly. Please see a notation added to figure 1.

---

## [Decision Letter · Decision Letter 1]

31 Dec 2021

High atherogenic risk concomitant with elevated HbA1c among persons with type 2 diabetes mellitus in North Ethiopia

PONE-D-21-17486R1

Dear Dr. Gebreyesus,

We’re pleased to inform you that your manuscript has been judged scientifically suitable for publication and will be formally accepted for publication once it meets all outstanding technical requirements.

Kind regards,

Xiao-Feng Yang, MD, PhD, FAHA

Academic Editor

PLOS ONE

Additional Editor Comments (optional):

Reviewers' comments:

Reviewer's Responses to Questions

**Comments to the Author**

1. If the authors have adequately addressed your comments raised in a previous round of review and you feel that this manuscript is now acceptable for publication, you may indicate that here to bypass the “Comments to the Author” section, enter your conflict of interest statement in the “Confidential to Editor” section, and submit your "Accept" recommendation.

Reviewer #1: All comments have been addressed

Reviewer #3: All comments have been addressed

2. Is the manuscript technically sound, and do the data support the conclusions?

Reviewer #1: Yes

Reviewer #3: Yes

3. Has the statistical analysis been performed appropriately and rigorously? 

Reviewer #1: Yes

Reviewer #3: I Don't Know

4. Have the authors made all data underlying the findings in their manuscript fully available?

Reviewer #1: Yes

Reviewer #3: Yes

5. Is the manuscript presented in an intelligible fashion and written in standard English?

Reviewer #1: Yes

Reviewer #3: Yes

6. Review Comments to the Author

Reviewer #1: Excellent! Since the authors have made significant revisions according to the comments raised by all reviewers, I am supportive of this study for publication in PONE.

Reviewer #3: Research Article titled “High atherogenic risk concomitant with elevated HbA1c among persons with type 2 diabetes mellitus in North Ethiopia” attempted to assess the glycemic status and CHD risk of T2DM using HbA1c and atherogenic index of plasma.

The strength of this paper is to analyze over four hundred T2DM patients in North Ethiopia to assess the CHD risk and glycemic status of T2DM by using HbA1c and AIP. The authors responded well to the reviewer’s comments, which could be accepted.

7. PLOS authors have the option to publish the peer review history of their article (what does this mean?). If published, this will include your full peer review and any attached files.

Reviewer #1: No

Reviewer #3: No

---

## [Editor Report · Acceptance letter]

24 Jan 2022

PONE-D-21-17486R1 

High atherogenic risk concomitant with elevated HbA1c among persons with type 2 diabetes mellitus in North Ethiopia 

Dear Dr. Gebreyesus:

I'm pleased to inform you that your manuscript has been deemed suitable for publication in PLOS ONE. Congratulations! Your manuscript is now with our production department. 

Kind regards, 

on behalf of

Dr. Xiao-Feng Yang 

Academic Editor

PLOS ONE